# Regulation of Hematopoietic Stem Cell Fate and Malignancy

**DOI:** 10.3390/ijms21134780

**Published:** 2020-07-06

**Authors:** Hee Jun Cho, Jungwoon Lee, Suk Ran Yoon, Hee Gu Lee, Haiyoung Jung

**Affiliations:** 1Immunotherapy Research Center, Korea Research Institute of Bioscience and Biotechnology (KRIBB), 125 Gwahak-ro, Yuseong-gu, Daejeon 34141, Korea; hjcho@kribb.re.kr (H.J.C.); sryoon@kribb.re.kr (S.R.Y.); 2Environmental Disease Research Center, Korea Research Institute of Bioscience and Biotechnology (KRIBB), 125 Gwahak-ro, Yuseong-gu, Daejeon 34141, Korea; jwlee821@kribb.re.kr; 3Department of Biomolecular Science, Korea University of Science and Technology (UST), 113 Gwahak-ro, Yuseong-gu, Daejeon 34113, Korea

**Keywords:** hematopoietic stem cell, aging, quiescence, self-renewal, differentiation, kinase inhibitor

## Abstract

The regulation of hematopoietic stem cell (HSC) fate decision, whether they keep quiescence, self-renew, or differentiate into blood lineage cells, is critical for maintaining the immune system throughout one’s lifetime. As HSCs are exposed to age-related stress, they gradually lose their self-renewal and regenerative capacity. Recently, many reports have implicated signaling pathways in the regulation of HSC fate determination and malignancies under aging stress or pathophysiological conditions. In this review, we focus on the current understanding of signaling pathways that regulate HSC fate including quiescence, self-renewal, and differentiation during aging, and additionally introduce pharmacological approaches to rescue defects of HSC fate determination or hematopoietic malignancies by kinase signaling pathways.

## 1. Introduction

Hematopoietic stem cells (HSCs) are quiescent and pluripotent cells that reside in bone marrow (BM) and continuously replenish blood cells throughout life [1]. HSCs can both self-renew and differentiate toward all blood lineages, and they maintain their homeostasis with low metabolic and cell cycle activity. The HSC pool is divided into two different subpopulations based on long-term reconstituting activity: long-term HSCs (LT-HSCs) and short-term HSCs (ST-HSCs), which can subsequently differentiate to multipotent progenitors (MPPs) that in turn differentiate into lymphoid or myeloid cells [2,3,4]. Dysregulation of HSC function can cause immunodeficiencies, anemia, hematopoietic failure, blood cancer, and death [5]. Under homeostatic conditions, HSCs retain the potential for long-term self-renewal and the capacity for subsequent reconstitution; however, severe hematopoietic stresses make HSCs lose this potential [6]. HSCs face a gradual decline in regenerative capacity and hematological pathologies with aging [7,8,9]. Aged HSCs show skewed myelopoiesis, functional decline, and pool expansion. In addition, HSC quiescence and concomitant attenuation of DNA repair causes DNA damage accumulation, which could induce pre-malignant mutations in aged HSCs [10].

In response to various signals, HSCs can be kept in quiescence, self-renew, or differentiate into lineage cells. These processes are regulated by various cellular signaling pathways, dysregulation of which results in defects of HSC function and hematopoiesis during aging. Elucidation of signaling pathways involved in HSC fate determination advances understanding of hematopoietic processes and may contribute to the development of efficient treatments for hematopoietic malignancies and age-related immune disorders. In this review, we introduce the signaling pathways that regulate HSC functions including quiescence, self-renewal, differentiation, and malignancy as well as recent approaches to overcoming defects in HSC fate determination or hematopoietic malignancies during aging.

## 2. General Features of Hematopoietic Stem Cell (HSC) Aging

Old bone marrow contains more HSCs than young bone marrow in both mice and humans [11,12,13]. This increase cannot compensate for the defects of aged HSCs and the aged HSC pool contained increased myeloid-dominant HSCs with a lower output of mature blood cells per HSC [14,15]. An increase in proliferation expanded the aged HSC subgroup and induced functional decline of HSCs [8]. Competitive transplantation assays have revealed a functional decline in the repopulation capacity of aged HSCs [1,16]. Hematopoiesis of aged HSCs produces more myeloid-biased compartments than hematopoiesis of young HSCs [1,17]. This is an autonomous process linked to upregulation of myeloid-specific gene expression in aged HSCs [18,19]. Single-cell transplantation assays also showed the dramatic increase of myeloid-restricted repopulating progenitors (MyRPs) within the phenotypic HSC compartment with age [20]. The accumulation of DNA damage has been observed in many reports during aging [10,21]. Aged HSCs show reduced self-renewal and regenerative capacities as well as impaired homing ability [22] (Figure 1).

## 3. Regulation of HSC Fate during Aging

### 3.1. Hematopoietic Stem Cell (HSC) Quiescence Regulation

Quiescence is the state of reversible arrest in the G_0_ stage of the cell cycle [23]. HSCs are kept in quiescence with low metabolic activity to maintain their numbers throughout life [24]. In response to hematopoietic stress, HSCs exit quiescence, proliferate, and differentiate to produce hematopoietic compartments. When quiescence of HSCs is disrupted, HSCs enter the cell cycle and are prematurely exhausted under hematopoietic stress [25]. HSC quiescence is critical for sustaining HSC pools throughout life and protects HSCs by minimizing replication-associated mutations in their genome [25,26]. HSC quiescence is regulated by a complex network of cell-intrinsic and -extrinsic factors [27]. Quiescent HSCs are activated by highly complex processes including epigenomic modulations, transcription, RNA processing, protein synthesis, DNA replication, mitochondrial biogenesis, and shifts in metabolic pathways [24]. Quiescent HSCs express low levels of DNA damage-related genes and HSC quiescence attenuates DNA repair or response pathways, which underlies the accumulation of DNA damage during aging [10]. However, activation of the cell cycle in HSCs can accelerate DNA-damage-driven aging, leading to its dysregulation in response to injury [28].

In the bone marrow, HSCs are located in the stem cell niche formed by supporting cells. The osteoblastic niche promotes the maintenance of quiescence for long-term repopulation of HSCs, whereas the vascular niche is thought to be where actively dividing stem or progenitor cells are located [25]. Depletion of osteoblasts reduces bone marrow cellularity and induces extramedullary hematopoiesis. Osteoblasts secrete factors such as stem cell factor (SCF), angiopoietin 1 (Ang-1), and thrombopoietin (TPO), which mediate interactions between HSCs and osteoblasts [29]. Receptor tyrosine kinase Tie2 on HSCs interacts with the Ang-1 ligand on osteoblasts and promotes HSC interactions with the extracellular matrix and cellular components of the niche to maintain quiescence and to enhance HSC survival by preventing cell division [30]. TPO is released from osteoblasts, and its receptor, myeloproliferative leukemia virus proto-oncogene (MPL), is expressed on HSCs and modulates HSC cell-cycle progression at the endosteal surface [31]. Several reports have raised controversial cellular function of osteoblasts in the bone marrow niche. As their ablation led to a loss of HSCs in the bone marrow; however, when CXCL12 was selectively deleted from osteoblasts, two different studies reported a loss of B-lymphoid progenitors rather than of HSCs, and conditional SCF knockout in osteoblasts did not affect the HSC number or function [29,32,33,34]. Another research group reported that osteoblast ablation did not decrease the numbers of LT-HSCs, but did impair their long-term reconstitution and self-renewal capacities [35]. Hypoxic niches are important for diminished HSC proliferation and quiescent HSCs situated in hypoxic environments [36]. Accumulation of DNA damage can also affect the systemic circulation of factors that influence stem cell and organism aging. DNA damage leads to the induction of cellular senescence, an irreversible state of cell cycle arrest, which blocks the proliferation of damaged or dysfunctional cells [24].

Regulation of HSC quiescence is orchestrated by multiple combinations of intrinsic factors including regulation of cell-cycle associated gene expression, chromatin modification, and microRNA-mediated gene expression [23]. Many cell cycle regulators are primarily involved in the regulation of HSC quiescence. Retinoblastoma protein (RB) is the main regulator of E2F transcription factor activity and mainly regulates the G1/S transition [37]. Deletion of all RB genes drives hyperproliferation in HSCs and induces impaired long-term HSC repopulation capacity [38]. Cyclin-dependent kinases (CDKs) and cyclin proteins negatively regulate RB activity and regulate the frequency of HSC division. Some CDK inhibitors such as p21^cip1/waf1^, p27^kip1^, and p57^kip2^ also regulate the HSC cell cycle [39]. Metabolic suppression is believed to maintain the low reactive oxygen species (ROS) production in HSCs; elevated ROS levels result in the loss of quiescence and self-renewal ability [40]. Elevated pyruvate dehydrogenase kinase (Pdk) expression leads to active suppression of the influx of glycolytic metabolites into mitochondria. Loss of both Pdk2 and Pdk4 attenuated HSC quiescence and repopulation capacity, and Pdk overexpression in glycolysis-defective HSCs restored quiescence and stem cell function [41]. Regulation of chromatin modification by deletion of polycomb group protein YY1 results in abolition of H3K27me3 modification and reduces quiescent HSCs [42]. Autophagy was increased in aged HSCs, which exhibited an overactive oxidative metabolism. Autophagy suppressed HSC metabolism by clearing active, healthy mitochondria to maintain quiescence, stemness, and regenerative capacity of aged HSCs [43]. A recent report has suggested that the Janus kinase/signal transducers and activators of transcription (JAK/STAT) signaling functions in stem cell proliferation and exhaustion during aging. Single-cell transcriptome analysis of HSCs showed that approximately 25% of p53-activated aged HSCs coexpressed cell cycle inhibitory and proliferative transcripts from JAK/STAT signaling, partially explaining the prolonged cell proliferation and HSC exhaustion [8]. Recent studies have shown that HSC quiescence is maintained and is tightly regulated during aging through complex interactions between intrinsic and extrinsic signals.

### 3.2. Regulation of HSC Self-Renewal and Differentiation

HSCs maintain their capacity for long-term self-renewal and the ability to generate all functional blood cells [44]. HSCs continuously provide differentiated progenitors while maintaining the HSC pool by precisely balancing self-renewal and differentiation [45]. HSC self-renewal involved in the maintenance or expansion of stem cell numbers following cellular division. Most HSCs are actively cycling during fetal life and old age, while HSCs in adulthood are often quiescent [15,46,47,48]. These phenomena were investigated by a serial transplantation assay, which showed that old HSCs had less self-renewal activity and generated smaller daughter clones in extended serial transplants than did their younger counterparts [14]. Slowly dividing HSCs in a steady state have improved self-renewal capacity, but enhanced cycling kinetics lead to loss of HSC self-renewal [49]. Thus, prolonged cell divisional kinetics is a representative phenotype that is associated with HSC maintenance throughout life.

These events occur in bone marrow niches, which include perivascular cells, endothelial cells, CXCL12-abundant reticular (CAR) cells, osteoblasts, neurons, and other cell types [50]. Perivascular stromal cells play a critical role in the regulation of HSCs and localize near both HSCs and endothelial cells [51]. In HSC niches, soluble and membrane-bound proteins participate in the regulation of HSC maintenance and self-renewal. CXCL12 is a chemokine for HSC homing or mobilization and a regulator of HSC quiescence [52]. The SCF and c-KIT receptors are important for supporting HSC survival, and partial inactivation of the c-KIT receptor reduces the self-renewal capacity and leads to loss of quiescence in HSCs [53]. TGF-β signaling is also important for the maintenance of quiescent HSCs. TGF-β receptor 1-deficient HSCs exhibit phenotypes in HSC function; however, TGF-β receptor 2 deficiency reduces the ability of HSCs to repopulate [54,55]. Conditional Smad4 deletion in HSCs impairs self-renewal and reduces the number of bone marrow cells [56]. Injection of IFN-α into mice stimulated quiescent HSCs to enter the cell cycle, and long-term exposure to IFN-α led to loss of HSC repopulation capacity [57]. Older adults frequently present with a systemic chronic low-grade inflammation that has been termed “inflammaging” [58]. Cytokines associated with inflammaging such as TNFα, IFNα, IFNγ, and IL-6 have been shown to reduce HSC self-renewal potential and result in myeloid-biased differentiation [59,60]. HSCs reside in a low oxygen bone marrow niche and maintain low ROS levels and long-term self-renewal activity [61,62].

Many transcription factors also regulate HSC function. For instance, PU.1 is correlated with the commitment of HSCs to myeloid, macrophage, and lymphoid lineages, and PU.1-knockdown HSCs showed impaired repopulation capacity because of the stimulation of the cell cycle, resulting in loss of HSCs [63]. FoxO3a-deficient HSCs showed a more severe reduction in repopulation capacity upon aging, and this HSC phenotype might be caused by aberrant accumulation of ROS [64]. Modulations of ROS and p53 activity by thioredoxin-interacting protein (TXNIP) may be implicated in HSC function during aging. TXNIP stabilizes and activate p53 via direct interaction, and regulates ROS levels in HSCs [65]. Rad21/cohesion-mediated NF-κB signaling to increase chromatin accessibility of NF-κB target genes in response to inflammation. Rad21 limits self-renewal of HSCs during aging and inflammation in a NF-κB-dependent manner [66]. In PTEN-like mitochondrial phosphatase (PTPMT1) knockout mice, hematopoietic failure was observed due to changes in the cell cycle and blockage of the differentiation of HSCs. The HSC pool was increased by ~40-fold in PTPMT1 knockout mice. Reintroduction of wild-type PTPMT1 restored the differentiation capabilities of PTPMT1 knockout HSCs [67]. Epigenomic profiling of HSCs revealed that aged HSCs exhibited reduced TGF-β signaling and broader H3K4me3 peaks across HSC identity and self-renewal genes. Aged HSCs showed increased DNA methylation at transcription factor binding sites associated with differentiation-promoting genes combined with a reduction at genes associated with HSC maintenance [68]. Aged HSCs express higher levels of RhoGTPase Cdc42, and this increase is related with HSC functional decline. Pharmacological inhibition of Cdc42 activity rejuvenated aged HSCs and rebalanced the differentiation of aged HSCs [69].

## 4. Regulation of HSC Fate and Malignancy

Aging of the hematological system causes anemia, reduced immunity, and increased incidence of hematological malignancies. The proteins kinases mediate intra- and intercellular signaling in various physiological and pathological conditions such as proliferation, differentiation, development, migration, apoptosis, inflammation, tumorigenesis, and immune-related disorders [70,71]. Protein kinases are implicated in the regulation of HSC function through phosphorylation reactions during self-renewal and differentiation. Aberrant regulation of the intracellular kinase pathways that underlie normal HSC self-renewal and differentiation resulted in impaired reconstitution capacity of HSCs and led to impaired hematopoiesis [72,73,74,75,76]. Here, we introduce some well-defined kinase signaling pathways and pharmacological applications using kinase inhibitors to regulate HSC fate and malignancies.

### 4.1. Phosphoinositide 3-Kinase (PI3K)

The PI3K family consists of three distinct subclasses and only the class I isoforms have been implicated in the regulation of hematopoiesis. PI3K class I has three catalytic subunits: p110α, p110β, and p110δ. The α and β isoforms are widely expressed in many tissues, whereas the γ and δ isoforms are highly restricted to hematopoietic cells. These catalytic subunits form heterodimers with a group a regulatory adapter molecules including p85α, p85β, p50α, p55α, and p55γ, and are frequently activated by protein tyrosine kinases [76,77,78]. The production of the lipid second messenger phosphatidylinositol (3, 4, 5) trisphosphate (PIP3) by PI3K is a crucial signal transduction mechanism in hematopoietic cells and affects multiple stages of hematopoietic development and mature hematopoietic cells [79,80]. Deficiency of the p85α subunit of PI3K complex in fetal liver HSCs results in reduced long-term engraftment and differentiation of HSCs [81]. Reduced PIP3 signaling resulted in the impaired HSC homeostasis and the development or function of T, B, and NK cells, myeloid mast cells, monocytes, granulocytes, and erythrocytes [82,83]. Combined deletion of p85α, p55α, and p50α resulted in a complete block in B lymphocyte development [76,84]. Introduction of a mutated, catalytically inactive p110δ (p110δ^D910A^) in the normal p110δ locus also resulted in a block in early B lymphocyte development [76,85]. Upon resolution of the stress, PI3K inactivation is required for the re-entry of HSCs into quiescence. Thus, PI3K activity in HSCs needs to be tuned into an appropriate window [5,79].

Wortmannin, a PI3K inhibitor, could cause defects in zebrafish hematopoiesis [86]. PI3K inhibitor, LY294002, completely abrogated the expansion during both neutrophil and eosinophil differentiation from human CD34^+^ HSCs and dramatically increased the percentage of apoptotic cells after extended culture [3]. A small molecule inhibitor of PI3K, NVP-BEZ235, is an imidazo[4–c]quinoline derivative that inhibits PI3K kinase activity by binding to the ATP-binding cleft of this enzyme [87]. NVP-BEZ235 decreased the ability of LSK Flk2^–^ cells to expand in a dose-dependent manner; these data demonstrated that stimulation of PI3K signaling was required for substantial HSC expansion in the culture system [88]. Many PI3K inhibitors were developed to regulate the activity of PI3K in hematopoietic cells as aberrant regulation of PI3K signaling is observed frequently in leukemia and is correlated with poor prognosis and drug resistance. Thus, PI3K is considered to be a promising target for therapy and PI3K inhibitors are mainly used for chemotherapeutic agents for hematological malignancies [76]. The PI3Kδ inhibitor, Idelalisib, has been approved for treating relapsed chronic lymphocytic leukemia (CLL), follicular B-cell non-Hodgkin lymphoma, and small lymphocytic Lymphoma [78,79]. Other PI3K inhibitors including S14161, the p110α-selective inhibitor AS702630, the p110β-selective inhibitor TGX-115, and the p110δ inhibitors IC87114 and CAL-10180 have been developed and regulate the proliferation and survival of leukemic blasts [76].

### 4.2. Protein Kinase B (PKB/AKT)

Protein kinase B (PKB/c-akt), a serine/threonine kinase, is an important effector of PI3K signaling [3,89]. Three highly homologous PKB isoforms have been described to be expressed in mammalian cells: PKBα, PKBβ, and PKBγ. PKB plays an important role in the regulation of cell survival and proliferation in various cell types [90]. PKBα and PKBβ are ubiquitously expressed and are present in greater abundance in hematopoietic cells, whereas PKBγ expression is most pronounced in the testes and brain; it can also be expressed in lesser amounts in the hematopoietic system [91,92,93,94,95,96]. PKBα is the most ubiquitously expressed isoform, and null mutant mice display growth retardation and increased levels of apoptosis. The mice are viable; however, their lifespan upon exposure to genotoxic stress is shorter compared with wild-type mice [3,91,93]. PKBβ is highly expressed in insulin-responsive tissues. PKBβ-deficient mice showed an insulin-resistant diabetic phenotype [97]. PKB is an important mediator of the PI3K signaling pathways in the regulation of hematopoiesis. PKBα/PKBβ double-knockout mice revealed defective long-term self-renewal capacity of HSCs. PKBα/PKBβ double deficient HSCs maintained the G0 phase of the cell cycle and showed defects in long-term hematopoiesis by enhanced quiescence. PKBα and PKBβ contributed to maintain a threshold of intracellular ROS in LSK cells [96]. The expression of myristoylated PKBα (constitutively activated PKBα) in HSCs revealed the transient expansion and increased cycling, associated with impaired engraftment. Finally, constitutive activation of PKB resulted in HSC exhaustion [98]. Constitutively active PKB resulted in enhanced neutrophil and monocyte development, whereas ectopic expression of dominant-negative PKB induced eosinophil development in vivo [3]. Constitutive activation of PKB resulted in the enhanced survival of acute myeloid leukemia (AML) and CLL cells [76].

Perturbing PKB activity during the recovery phase of HSC after stress led to altered HSC activity. Perifosine, a PKB inhibitor, is an alkylphospholipid that is thought to be incorporated into cell membranes, and limits the accessibility of membrane signaling domains for PKB, and subsequently blocks PKB activation via phosphorylation. Perifosine rescued the engraftment defect of CD81 deficient HSCs [99]. Perifosine induces apoptosis in multidrug-resistant human T-ALL cells and primary AML cells, but does not affect normal CD34^+^ hematopoietic progenitor cells [76,100,101]. A potent and selective inhibitor of Akt1 and Akt2 (AKTi-1/2; naphthyridinone) was prepared by optimization of leads identified by a high-throughput screen for inhibitors of purified activated Akt1, Akt2, and Akt3 kinases. AKTi-1/2 is not competitive with ATP and its inhibitory activity requires the presence of the PH domain [102]. AKTi-1/2 promotes quiescence and enhances engraftment of human UCB CD34^+^ cells in immunodeficient mice. This study may facilitate clinical strategies that can enhance the engraftment of human UCB HSPCs [103]. Phosphatidylinositol ether lipid analogs (PIAs), SH-5 and SH-6, were as effective as LY294002 in decreasing the amount of the PKB phosphorylated forms and inhibited proliferation and sensitization of HL60 cells to chemotherapeutic agents in concentrations that did not affect proliferation of normal hematopoietic progenitors [104]. The PKB inhibitor triciribine, a purine analog, has been demonstrated to interact with the PH domain of PKB and can thus prevent the association of PKB with PI(3,4,5)P3. Triciribine showed a significant effect on CD49f^+^ cell (a HSC subset) survival and, among the survivors, cell division was significantly delayed [105]. In T-ALL cell lines, triciribine has been demonstrated to induce cell cycle arrest and apoptosis [106].

### 4.3. Mammalian Target of Rapamycin (mTOR)

The mammalian target of rapamycin (mTOR) is a serine/threonine kinase. It is activated by PI3K, PDK1, and PKB in response to nutrients, growth factors, and intracellular energy status [107,108]. mTOR is a part of the phosphatidylinositol kinase-related kinase subfamily. mTOR forms two distinct multi-protein complexes: rapamycin-sensitive mTOR complex 1 (mTORC1) and rapamycin-insensitive mTOR complex 2 (mTORC2) [107,109]. mTORC1 has mTOR, Raptor, PRAS40, and mLST8, and functions as the key regulator of protein synthesis and cell growth. mTORC2 includes mTOR, Rictor, Sin1, and mLST8, and mainly functions to regulate cell survival/proliferation and actin cytoskeleton organization through the phosphorylation of PKB [109]. mTOR activation leads to phosphorylation of S6K1 and 4E-BP to regulate protein synthesis, cell growth, and metabolism, and to cell survival via phosphorylating PKB on Ser473 [110]. In the hematopoietic system, mTOR regulates proliferation and differentiation of megakaryocytes and dendritic cell and hyper-activation of mTOR by the deletion of negative regulators, thus resulting in long-term HSC exhaustion [75,111,112,113]. Gene targeting of mTOR in embryonic stem cells results in early embryonic lethality [114]. Under normally activating conditions, T cells lacking mTOR differentiated into Foxp3^+^ regulatory T cells. mTOR signaling regulates decisions between effector and regulatory T cell lineage commitment [115]. The physiological roles of mTOR in hematopoiesis and HSC function were examined using a hematopoietic-specific inducible mouse knockout model. mTOR deficiency caused bone marrow failure, a reduction in blood lineage production, and impaired HSC engraftment [75]. Conditional deletion of TSC1 in mice, resulting in activation of mTOR, has been demonstrated to enhance the percentage of cycling HSCs and to reduce the self-renewal capacity of HSCs in serial transplantation assays [113].

Rapamycin, an inhibitor of mTOR, rescued both the disease phenotype and the HSC phenotype of mice with a conditional deletion of phosphatase and tensin homolog (PTEN). Rapamycin partially rescued cobblestone formation and colony formation by myristoylated PKB BM [98]. PTEN deletion promoted HSC proliferation; however, this led to HSC depletion via a cell-autonomous mechanism, preventing these cells from stably reconstituting irradiated mice. In contrast to leukemia-initiating cells, HSCs were unable to maintain themselves without PTEN. Rapamycin not only depleted leukemia-initiating cells, but also restored normal HSC function [116]. Many rapamycin analogs were developed to regulate mTOR activity such as RAD001 (everolimus), CCI-779 (temsirolimus), AP23573 (deforolimus), and RAD001, and they inhibited the mTORC1 complex through association with FKBP-12, which abrogates the association of Raptor with mTOR [76,117,118,119]. The efficacy of these compounds has been examined in various preclinical and clinical studies for hematological malignancies. ATP-competitive mTOR inhibitor, PP242 inhibits mTORC 1/2 and efficiently reduced the development of leukemia in mice transplanted with primary ALL blasts or preleukemic thymocytes overexpressing PKB, while inducing less adverse effects on the function of normal lymphocytes [120,121]. mTORC1/2 dual inhibition by KU-63794 or AZD8055 suppresses day-14 cobblestone area-forming cells (CAFCs), but not day-35 CAFCs human HPSCs. CD34^+^ human hematopoietic cells appear to be resistant to AZD8055-induced toxicity. From these results, they suggest that like in mouse HSPCs, dual mTORC1/2 inhibition may have differential effects on human HSCs versus HPCs [122].

### 4.4. Glycogen Synthase Kinase-3 (GSK-3)

Glycogen synthase kinase-3 (GSK-3) is a constitutively active serine/threonine kinase [123], originally identified as inactivating glycogen synthase [124]. The GSK-3 gene family consists of two highly conserved kinases: GSK-3α and GSK-3β. GSK-3α and GSK-3β share 98% sequence identity in their kinase domains but 36% identity in their carboxyl terminus [125]. GSK-3 isoforms exhibit tissue-specific physiologically important functions that are sometimes different. Most studies have focused on GSK-3β; however, GSK-3α has also been identified as a key target in AML [126,127]. GSK-3 is involved in the regulation of HSC function by either the ectopic expression of key upstream regulators or by presentation of ligands in vitro. GSK-3 regulates the Wnt, Hedgehog, and Notch pathways [128]. Loss of functional GSK-3 activity in the bone marrow (BM) transiently expanded HSCs in a β-catenin-dependent fashion, documenting the role for Wnt/β-catenin signaling in homeostasis [73]. In assays of long-term HSC function, loss of GSK-3 progressively depleted HSCs through the activation of mTOR. HSC depletion was rescued by mTOR inhibition and exacerbated by β-catenin knockout. These suggest that GSK-3 regulates both Wnt and mTOR signaling in mouse HSCs, with these pathways promoting HSC self-renewal and lineage commitment, respectively [73,127].

GSK-3 inhibitor (CHIR-911) enhances cytopenic recovery and HSC repopulation. In vivo administration of CHIR-911 directly promotes proliferation and the overall expansion of primitive cells to enhance hematopoietic repopulating activity and increases the hematopoietic repopulation in recipients transplanted with mouse or human HSCs. CHIR-911 treatment shortened the neutrophil and megakaryocyte recovery period, improved survival of transplanted mice and sustained enhanced long-term HSC repopulation [128]. GSK-3 inhibitor lithium-treated mice showed an increase in HSCs/HPCs, as detected by LSK markers or by the detection of the SLAM marker (CD150^+^CD48^–^) characteristic of HSCs [73,129,130]. The selective GSK-3 inhibitor 6-bromoindirubin 3′-oxime (6BIO) also increased the number of LSK cells after two weeks and the hematopoietic progenitors in CFU-S_12_ analysis [73,131]. AR-GSK-3 inhibitor, A014418, has high specificity for GSK-3, inhibits GSK-3 in an ATP-competitive manner, and also increased hematopoietic colony formation ex vivo [73,132]

### 4.5. p38 Mitogen-Activated Protein Kinase (p38)

The p38 mitogen-activated protein kinase (p38) belongs to the MAPK family of signal transduction kinases [133]. p38 has four isoforms: p38α, p38β, p38γ, and p38δ. p38α is predominantly expressed and increased in LT-HSCs [1,134]. It is activated in a sequential order (mitogen-activated or extracellular signal-regulated kinase kinase-MAPK kinase 3/6-p38) to regulate various cellular processes such as differentiation, cell cycle arrest/senescence, and apoptosis in a cell type specific manner [133,135]. p38 also plays an important role in the regulation of hematopoiesis, particularly erythropoiesis and granulopoiesis [134]. Erythropoietin (EPO), interleukin-3 (IL-3), granulocyte colony stimulating factor (G-CSF), and thrombopoietin (TPO) stimulate p38 activity, which induces HSCs/HPCs proliferation and differentiation [134,136,137,138]. Sorted mouse BM LSK cells exhibited selective activation of p38 and the activation of p38 was associated with a significant reduction in HSCs and induction of apoptosis and cellular senescence in LSK+ cells and their progeny [135]. p38 activation in various pathological conditions or during cellular aging via elevated ROS, results in HSC defects [1]. Activation of p38 MAPK in response to increasing levels of ROS limits the lifespan of HSCs in vivo. In ATM deficient mice, elevation of ROS levels induces HSC specific phosphorylation of p38 MAPK accompanied by a defect in the maintenance of HSC quiescence [139]. p38 MAPK activation is important in the process of regulating the growth inhibitory signals of TNF-α, TGF-β, and interferons on human hematopoiesis. p38 is overactivated in myelodysplasia BMs and regulates HSC apoptosis [140].

SCIO-469 is an ATP-competitive p38 inhibitor and selectively inhibits the activity of p38α. SCIO-469 could block cytokine induced phosphorylation of p38 and its downstream effector in primary hematopoietic progenitors and in MDS1, a cell line derived from a myelodysplastic syndrome (MDS) patient. SCIO-469 treatment led to a significant decrease in apoptotic CD34^+^ cells. Colony forming assays revealed that MDS CD34^+^ hematopoietic progenitors induced more erythroid and myeloid colony formation in vitro after treatment with SCIO-469 at low doses [140]. A p38 inhibitor SB203580 rescued ROS-induced defects in HSC repopulating capacity and in the maintenance of HSC quiescence, and extended the lifespan of HSCs from wild-type mice in serial transplantation experiments [139]. Cylindromatosis (CYLD), a tumor suppressor gene and negative regulator of NF-κB signaling with deubiquitinase activity, is highly expressed in dormant HSCs (dHSCs). Deletion of the catalytic domain of CYLD induced dHSCs to exit quiescence and abrogated their repopulation and self-renewal potential. This phenotype depends on the activation of the p38 pathway. LY2228820 and BIRB 796, p38 inhibitors [141,142,143], both at least partially restored HSC quiescence in CYLDex7/8 mutant HSCs, but did not alter the cell-cycle distribution of control cells [144]. The activation of p38 induces defects of HSC function such as lineage skewing, decrease in engraftment, an increase in reactive oxygen species, and loss of Cdc42 polarity. A cell-penetrating peptide (CPP)-conjugated peptide (TAT-TN13) derived from the TXNIP-p38 interaction motif inhibited p38 activity via this docking interaction. A new p38 inhibitor TAT-TN13 selectively inhibited the p38α isoform and dramatically reduced p38 activity in old HSCs comparable to that of SB203580 treatment. TAT-TN13 treatment reduced ROS levels, returned depolarized old HSCs to polarized HSCs, and increased the homing ability of old HSCs after short-term transplantation. TAT-TN13-treated old HSCs or 12-month-old TXNIP deficient HSCs exhibited restoration of aged phenotypes of HSCs comparable to SB203580 [1].

### 4.6. Other Kinases and Their Inhibitors

Janus kinase 1(JAK1) is a critical effector of pro-inflammatory cytokine signaling and immune function and abnormal JAK1 activity has been linked to immunological and neoplastic diseases [74]. Conditional JAK1 loss in mice had reduced white blood cell count and frequency of HSCs and induced HSC quiescence and myeloid skewing. JAK1-deficient HSCs exhibit decreased competitive repopulation and are unable to rescue hematopoiesis from myelosuppression (5-fluorouracil (5-FU) treatment). JAK1-deficient HSCs have an intrinsic defect in IL-3 responsiveness and the ability of JAK1-deficient LSKs to form colonies in the presence of IL-3 shows a 50% reduction in colony numbers. JAK1 specific inhibitor GLPG0634 inhibited the expansion of wild-type cells ex vivo by the addition of IL-3 and IFN-α [74].

Phosphoinositide dependent kinase-1 (PDK1) plays an important role in the biological function of B cells, T cells, and platelets. PDK1 was investigated as a therapeutic target for cancers including leukemia [145]. PDK1 phosphorylates at threonine 308 of PKB and is also known as the AGC kinase; it is an important downstream molecule of the PI3K signaling pathway [146]. PDK1 phosphorylates its targets to regulate cell metabolism, growth, survival, and anti-apoptosis [147,148,149]. PDK1 deletion severely impaired the repopulating potential of HSCs in murine E15.5 fetal liver (FL). PDK1 deficient FL HSCs showed enhanced apoptosis and proliferation ability. PDK1 maintains FL hematopoiesis by regulating HSC apoptosis and cell cycle by the Akt-FOXO signaling pathway [150].

Liver kinase B1 (LKB1), also known as serine/threonine kinase 11 (STK11), is a tumor suppressor and a conserved regulator of cellular energy metabolism in eukaryotic cells. LKB1 phosphorylates AMP-activated protein kinase (AMPK) and AMPK-related kinases [151]. AMPK activates the tuberous sclerosis complex, which inhibits the mTOR complex 1 (mTORC1), [152,153] and AMPK also inactivates mTORC1 by phosphorylating Raptor [154]. LKB1-deficient mice showed increased HSC division, rapid HSC depletion, a marked reduction of HSC repopulating potential and pancytopenia. HSC depletion did not depend on mTOR activation or oxidative stress. LKB1-deficient HSCs, but not AMPK-deficient HSCs, exhibited defects in centrosomes and mitotic spindles in culture, and became aneuploid. LKB1 is required for HSC maintenance through AMPK-dependent and AMPK-independent mechanisms [155,156].

Proviral insertion in murine lymphomas protein kinases are a small family of constitutively active, highly conserved oncogenic serine/threonine kinases that have three members: PIM1, PIM2, and PIM3 [157,158,159]. They share more than 60% homology at the amino acid level and their functions are overlapped in lymphomagenesis [160,161]. PIM1 transgenic mice (Pim1-Tx) overexpressing human PIM1 showed increased numbers of LSK hematopoietic stem/progenitor cells and CAFCs, higher BrdU incorporation in long-term HSC population, and a better ability to reconstitute in recipient mice. Pim1 single knockout (KO) mice showed impaired long-term hematopoietic repopulating capacity in secondary and competitive transplantations. However, these defects were not observed in HSCs from Pim2 or Pim3 KO mice [161].

Protein kinases regulate HSC function during the self-renewal and differentiation through phosphorylation reactions. Aberrant regulation of the intracellular kinase pathways reduces the function of HSCs and leads to impaired hematopoiesis (Figure 2).

## 5. Conclusions

In this review, we have introduced many of the signaling pathways regulating the fate of HSCs including quiescence, self-renewal, and differentiation during aging, and the malignancy of HSCs. These processes are regulated by various cellular signaling pathways, and dysregulation of these signaling pathways result in defects of HSC function and hematopoietic malignancies. Recently, many reports have implicated phosphorylation reactions in kinase signaling pathways in the regulation of HSC function during normal and malignant hematopoiesis. They have also suggested that specific inhibitors of kinases can reverse defects of HSC function under stress or malignant conditions. Although we have described the defined signaling pathways, and pharmacological regulation in HSC fate determination and malignancies, it does not explain all the aspects of HSC biology related to aging and remains elusive. However, understanding the regulation of HSC fate and malignancy by intrinsic- and extrinsic signaling pathways and pharmacological inhibitors may be a critical issue for the treatment of hematological malignancies or aging-related immune-disorders.

## Figures and Tables

**Figure 1 ijms-21-04780-f001:**
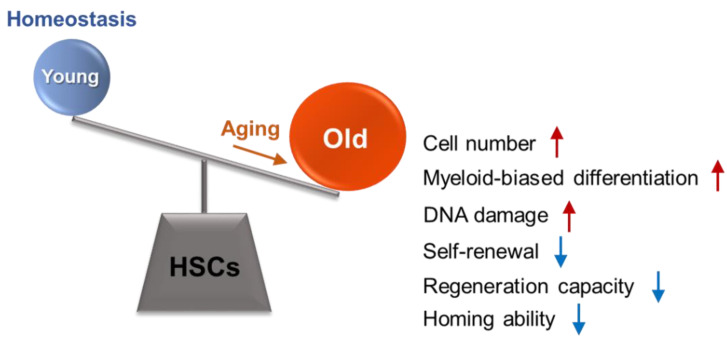
General phenotypes of aged hematopoietic stem cells (HSCs). Aged HSCs show increased cell number, myeloid-biased differentiation, DNA damage accumulation, reduced self-renewal, reduced regeneration capacity, and reduced homing ability compared with young HSCs.

**Figure 2 ijms-21-04780-f002:**
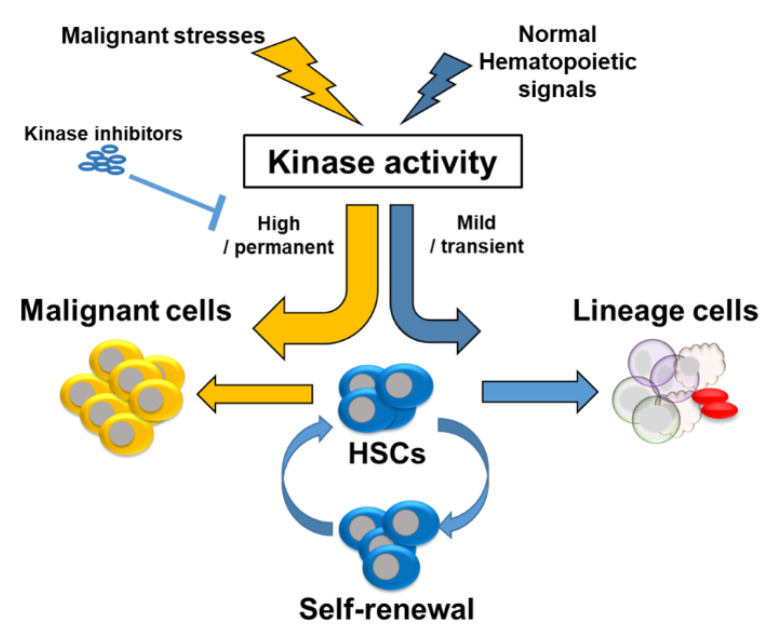
Fate decision of HSCs by kinase signaling. Normal hematopoietic signals moderately activate kinase signaling pathways of HSCs to self-renew or to differentiate into lineage cells. However, malignant stresses strongly or permanently activate kinase signaling pathways of HSCs and subsequently induce hematopoietic malignancies.

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
