# Peer review of "Regulation of Hematopoietic Stem Cell Fate and Malignancy"

_ijms, 2020, doi:10.3390/ijms21134780_

Round 1
Reviewer 1 Report
Cho et al present the many of signaling pathways regulating the fate of HSCs, including quiescence, self-renewal and differentiation, and the malignancy of HSCs during aging. These processes are regulated by various cellular signaling pathways, and dysregulation of these signaling pathways result in defects of HSC function and hematopoietic malignancies. The paper is of good quality, but it needs professional English proofing.
Author Response
We appreciate the reviewer’s comments. We have fully edited English language again.
Reviewer 2 Report
The manuscript entitled “Regulation of Hematopoietic Stem Cell Fate and Malignancy during Aging” by Hee Jun Cho et al. reviews aspects involved in HSC fate decision and how they change upon aging.
Overall, the manuscript is referencing mainly to other reviews than to experimental work and the referenced papers are relatively old (with few exceptions). Moreover, while in the general introduction about features of HSC aging the authors offer a rather old overview of some phenotypic and functional alterations of aged HSCs, in the rest of the review there is no clear definition of aged versus young stem cell features and all reported data applies mainly to young HSCs. Therefore, most of the review doesn’t focus on characteristics of aged stem cells.
Major points:
- Lines 51-53: the references to support the statement should include the primary research works, not only other reviews
- Lines 53-54: “Aged HSCs are less quiescent and are divided more frequently than young HSCs”: the sentence is unclear or wrongly phrased. The reference to another review is not appropriate, should be to the primary research paper. Also, the topic is quite debated and I believe that the consensus is not that aged HSCs proliferate more than young.
- The section 3.1 HSC quiescence regulation deals with quiescence but not specifically with aging. The reference to what happens upon aging is not clearly described. It consists of a compilation of extrinsic and intrinsic regulators described for the young hematopoietic system. References are overall quite old or other reviews.
- Lines 112-113 “Most HSCs are actively cycling during fetal life and old age”: please reference the experimental work showing these data for old HSCs
- Lines 122-127: this sentence is contrasting with what written above in lines 79-85. Please clarify.
- In the section 2. Regulation of HSC self-renewal and differentiation, the final parts (Lines 128-142) dealing with soluble factors and transcription factors are very brief and there is no clear reference to what changes with aging.
- The sections 4.1. Phosphoinositide 3-kinase (PI3K) and 4.2. Protein kinase B (PKB/AKT) do not address any specific data about aged HSCs.
- Lines 385-386: “In this review, we have introduced many of signaling pathways regulating the fate of HSCs including quiescence, self-renewal and differentiation, and the malignancy of HSCs during aging.” The sentence should be rephrased and in general the specific phenotypes related to aged HSCs are not clearly described all over the manuscript, beside the initial part dealing with general features of aged HSCs.
- The cartoon scheme is too general and might be mis-leading. What does “aging stresses” exactly indicate? Which kinase activity is increased in HSCs with aging? Aging does not always lead to malignancy. Aging and malignancy should be distinct.
Author Response
We appreciate the reviewer’s comments. We have fully edited English language again. We have added more articles related with aging phenotype of HSCs and we have tried to express the differences between young and aged HSCs in phenotypic and functional features. Please check a new manuscript version and modified manuscript was clearly highlighted by the red colored text.
The manuscript entitled “Regulation of Hematopoietic Stem Cell Fate and Malignancy during Aging” by Hee Jun Cho et al. reviews aspects involved in HSC fate decision and how they change upon aging. Overall, the manuscript is referencing mainly to other reviews than to experimental work and the referenced papers are relatively old (with few exceptions). Moreover, while in the general introduction about features of HSC aging the authors offer a rather old overview of some phenotypic and functional alterations of aged HSCs, in the rest of the review there is no clear definition of aged versus young stem cell features and all reported data applies mainly to young HSCs. Therefore, most of the review doesn’t focus on characteristics of aged stem cells.
Response: We appreciate the reviewer’s comments. We have fully edited English language again. We have added more articles related with aging phenotype of HSCs and we have tried to express the differences between young and aged HSCs in phenotypic and functional features. Please check a new manuscript version and modified manuscript was clearly highlighted by the red colored text.
Major points:
- Lines 51-53: the references to support the statement should include the primary research works, not only other reviews
Response: We have added more primary research works to support the statement.
- Lines 53-54: “Aged HSCs are less quiescent and are divided more frequently than young HSCs”: the sentence is unclear or wrongly phrased. The reference to another review is not appropriate, should be to the primary research paper. Also, the topic is quite debated and I believe that the consensus is not that aged HSCs proliferate more than young.
Response: We have referred to Irving L. Weissman group’s report. They found that aged human HSC increase in frequency, are less quiescent, and exhibit myeloid-biased differentiation potential compared with young HSC (PNAS 108: 20012-20017, 2011). This topic may be quite debated so, we have deleted or changed the sentence and added more recent articles related with HSC aging phenotype. We have also edited Figure 1.
- The section 3.1 HSC quiescence regulation deals with quiescence but not specifically with aging. The reference to what happens upon aging is not clearly described. It consists of a compilation of extrinsic and intrinsic regulators described for the young hematopoietic system. References are overall quite old or other reviews.
Response: We have added some more recent articles related with HSC aging.
- Lines 112-113 “Most HSCs are actively cycling during fetal life and old age”: please reference the experimental work showing these data for old HSCs
Response: We have added 3 references. S. J. Morrison et al. have described that HSCs in young and middle–aged mice rarely were in the S/G2/M phases of the cell cycle, but HSCs in old mice were frequently in cycle (Nature Medicine 2:1011-1016, 1996).
- Lines 122-127: this sentence is contrasting with what written above in lines 79-85. Please clarify.
Response: Lines 122-127 were combined into the end of lines 79-85. We introduced this sentence as a controversial results for the cellular function of osteoblasts in bone marrow niche.
- In the section 2. Regulation of HSC self-renewal and differentiation, the final parts (Lines 128-142) dealing with soluble factors and transcription factors are very brief and there is no clear reference to what changes with aging.
Response: We have added more primary research works related with HSC aging.
- The sections 4.1. Phosphoinositide 3-kinase (PI3K) and 4.2. Protein kinase B (PKB/AKT) do not address any specific data about aged HSCs.
Response: In the section 4, we would like to focus on the regulation of HSC malignancy because aging of the hematological system can increase incidence of hematological malignancies. We rephrased the sentence “In this review, we have introduced many of signaling pathways regulating the fate of HSCs including quiescence, self-renewal and differentiation during aging, and the malignancy of HSCs.” to “In this review, we have introduced many of signaling pathways regulating the fate of HSCs including quiescence, self-renewal and differentiation, and the malignancy of HSCs during aging.”.
- Lines 385-386: “In this review, we have introduced many of signaling pathways regulating the fate of HSCs including quiescence, self-renewal and differentiation, and the malignancy of HSCs during aging.” The sentence should be rephrased and in general the specific phenotypes related to aged HSCs are not clearly described all over the manuscript, beside the initial part dealing with general features of aged HSCs.
Response: To describe the phenotypic and functional differences between young and aged HSCs, we have added more references related with HSC aging in the sections 2 and 3. In the section 4, we would like to focus on the regulation of HSC malignancy because aging of the hematological system can increase incidence of hematological malignancies. We rephrased the sentence “In this review, we have introduced many of signaling pathways regulating the fate of HSCs including quiescence, self-renewal and differentiation during aging, and the malignancy of HSCs.” to “In this review, we have introduced many of signaling pathways regulating the fate of HSCs including quiescence, self-renewal and differentiation, and the malignancy of HSCs during aging.”. We also rephrased the manuscript title and abstract. Please check a new manuscript version and modified manuscript was clearly highlighted by the red colored text.
- The cartoon scheme is too general and might be mis-leading. What does “aging stresses” exactly indicate? Which kinase activity is increased in HSCs with aging? Aging does not always lead to malignancy. Aging and malignancy should be distinct.
Response: In the section 4 and Figure 2, we tried to emphasize the relationship between HSC aging and malignancy because aging of the hematological system can increase incidence of hematological malignancies. Aging stresses of HSCs may contain ROS, DNA damage or mutation, epigenetic changes, etc. Please refer to reference 11. We edited Figure 2 and figure legend. We removed “aging stresses” from figure and legend.